# Faster progression to multiple sclerosis disability is linked to neuronal pathways associated with neurodegeneration: An ethnicity study

Gil Harari[1,2]*, Michael Gurevich[2,3], Mark Dolev[2,3], Rina Zilkha Falb[2], Anat Achiron[2,3]

**1** School of Public Health, University of Haifa, Haifa, Israel, **2** Multiple Sclerosis Center, Sheba Medical Center, Ramat Gan, Israel, **3** Sackler School of Medicine, Tel-Aviv University, Tel Aviv, Israel

* gil@medistat.co.il

**Data Availability Statement:** All relevant data are within the manuscript and its Supporting Information files.

## Abstract

Although the causes of multiple sclerosis are largely unknown, genetic and environmental components play an important role. Geographic distribution, varying with latitude, reflects both genetic and environmental influences. We conducted a retrospective exploratory observational study to characterize the disability progression of 2396 Jewish patients with relapsing-remitting multiple sclerosis, followed at the Sheba Multiple Sclerosis Center, Tel-Aviv, Israel; 188 patients who originated in Iraq and 2207 patients who originated in northern Europe. Peripheral blood microarray gene expression analysis was performed in a sub-group of patients to identify molecular pathways associated with faster disability progression. During a follow-up period of 18.8 and 19.8 years, respectively, 51.6% of patients with an Iraqi origin progressed to moderate disability defined as expanded disability status scale (EDSS) score of 3.0 to 5.5, compared to 44.2% of patients with a northern European origin (odds ratio 1.347, 95% CI 1.0–1.815, p = 0.049). An Iraqi origin was associated with increased risk of progression to moderate disability adjusted for sex, disease duration, age at onset, and treatment with immunomodulatory drugs (hazard ratio 1.323; 95% CI, 1.049–1.668, p = 0.02), but not to severe disability defined as EDSS score > = 6.0 (i.e., walking aids are required for a distance of 100 meters, (hazard ratio 1.311; 95% CI, 0.918–1.874, p = 0.136). Gene expression analysis disclosed 98 differentially expressed genes (79 over-expressed and 19 under-expressed) between relapsing-remitting multiple sclerosis patients of Iraqi origin (N = 17) and northern European (N = 34) origin. Interestingly, this gene expression was enriched with genes related to neuronal pathways associated with morphology of axons, branching of neurites, proliferation of neocortical neurons, and formation of myelin sheath, suggesting an augmented process of neurodegeneration in relapsing-remitting multiple sclerosis patients with an Iraqi origin. The study results suggest that relapsing-remitting multiple sclerosis patients with an Iraqi origin progress faster to disability possibly due to an enhanced process of neurodegeneration.

**Funding:** The authors received no specific funding for this work.

**Competing interests:** Anat Achiron served on the scientific advisory board and/or received speaker honoraria/consulting from Biogen, BMS, Merck, Novartis, Roche and Sanofi, received grant/research support from Biogen, BMS, Merck, Roche and Sanofi and received institutional support from Sackler School of Medicine, Tel-Aviv University for Autoimmune Diseases research. All other authors have no competing interests. This does not alter our adherence to PLOS ONE policies on sharing data and materials

## Introduction

Multiple sclerosis (MS) is an inflammatory disease of the central nervous system affecting young adults, leading to progressive neurologic disability overtime [1]. The causes of MS are largely unknown, but an interplay of genetic and environmental components plays an important role [2]. Among these components, geographic distribution, varying with latitude, has an important role as it reflects both genetic and environmental influences. MS is common in northern Europe, north America, and Australia, but rare in the Asia, Africa and South America. Therefore, immigrant studies are of importance to better understand these disease-related causative factors, as they provide an opportunity to better isolate the genetic effect on the background of environmental changes. The state Israel provides a possibility to study MS-related immigrant effects as profound demographic changes have taken place in the Israeli society as a result of successive waves of immigration from all over the world. Over the last 70 years, gradually, Jewish people from different countries of origin became assimilated and each immigrant group, with its own unique characteristics, contributed differently to the MS population.

We have observed in our clinical practice that patients with relapsing-remitting MS who originated from Iraq have an aggressive disease course with relatively rapid progression to disability.

In the current study we aimed to further evaluate the progression rate of relapsing-remitting MS patients who originated from Iraq. We assessed the effects of an Iraqi origin on MS disability progression as compared to relapsing-remitting MS patients who originated from northern Europe.

## Methods

### Study design and participants

This study is a retrospective exploratory observational study that compared the rate of neurological disability progression in Jewish relapsing-remitting MS patients who either originated from Iraq or had a northern European origin. Peripheral blood gene expression signatures were compared is a sub-group of patients to identify transcripts related to origin-associated disease progression. We obtained demographic and clinical data including age at onset, disability assessment overtime as measured by the Expanded Disability Status Scale (EDSS) [3], and treatment duration of immunomodulatory drugs (IMDs), from the Sheba Medical Center Multiple Sclerosis (SMCMS) data registry. The SMCMS registry is certified by the Israeli Ministry of Justice (Registry number 597247) and is increasingly being used in research [4,5]. Information about ethnic background is routinely collected and index cases were included in this study according to the following inclusion criteria: 1) fulfilled the 2017 McDonald diagnostic criteria for MS [6] supported by MRI examination with typical demyelinating lesions [7]; 2) Iraqi-born Jewish patients or Israeli-born Jewish patients with both parents born in Iraq. For comparison we obtained data of (1) Ashkenazi Northern Europe-born patients or Israeli-born Jewish patients with both parents from Ashkenazi origin and born in northern Europe. It is of note that Iraqi originated Jews are non-Ashkenazi (defined as Sephardic or Orientals), and Northern Europe origin Jews are defined as Ashkenazi. Mixed patients (one parent from Northern Europe, one parent from Iraq) were not included in the study.

The primary outcome event was disability progression defined as the time to reach irreversible moderate and severe disability. Moderate disability was defined as an EDSS disability score of at least 3.0 and up to 5.5; severe disability was defined as an EDSS disability score equal or greater than 6.0 (i.e., walking aids are required for a distance of 100 meters). Disability was defined as irreversible when a patient had a given score for at least six consecutive months.

The study was approved by the Sheba Medical Center Institutional Review and Ethical Board; all patients gave written informed consent for the blood microarray study.

## Statistical analysis

The $\chi2$ distribution and Fisher tests were used to assess significance in all instances, except for when looking at age data, where a 2-tailed Student t test was used to assess significance. Country of origin risk was calculated based on a modification of the maximum likelihood approach. For categorical variables summary tables are presented giving sample size, absolute and relative frequency by study origin. For continuous variables summary tables are presented giving sample size, median, and 25% and 75% percentiles by origin. All tests used are two-tailed, and a p value of 5% or less is considered statistically significant. The data was analyzed using the SAS® version 9.4 (SAS Institute, Cary North Carolina). Chi-square test was applied for testing the statistical significance of the difference in percent of patients with confirmed moderate and severe disability progression between the two populations. The hazards ratio and odds ratios for confirmed moderate and severe disability progression between patients who originated in Iraq and those who originated in Northern Europe were estimated via the Cox regression model and logistic regression with adjustment for age, sex, disease duration, and treatment with IMDs.

## Microarray preparation

Total RNA was extracted from peripheral blood mononuclear cells using Trizol (Invitrogen, USA) and Phase-Look-Gel columns (Eppendorf, Germany). RNA quality was determined by BioRad Experion automatic electrophoresis station. cDNA was synthesized using the One-Cycle cDNA Synthesis Kit, transcribed by GeneChip IVT Labeling Kit (Affymetrix, Inc. CA.), hybridized to HGU133A-2 microarrays, washed in a GeneChip Fluidics Station 450, and scanned on GeneArray-TM scanner (G2500A, Hewlett Packard) according to Affymetrix Inc protocol.

## Microarray data pretreatment and statistical analysis

Microarray data was normalized by R Bioconductor Packages (https://www.bioconductor.org) as follows: a) arrays were normalized using single-sample microarray normalization; b) batch effect was treated by Combat SVA package [8]. Partek Genomics Software (https://www.partek.com/partek-genomics-suite) was used for statistical analysis. Differentially expressed genes (DEGs) were determined using p-value cut off <0.05 after False Discovery Rate (FDR) correction for multiple comparisons. DEGs were applied for functional analysis using QIAGEN's Ingenuity Pathway Analysis (IPA®, QIAGEN Redwood City, www.qiagen.com/ingenuity).

## Results

### Cohort characteristics

Clinical and demographic cohort characteristics are presented in Table 1. There were no significant changes between relapsing-remitting MS patients who originated in Iraq and patients who originated in Northern Europe in relation to age at onset and disability at onset. The prevalence of women was higher in patients who originated in Northern Europe. The follow-up time was similar, and the percent of patients treated with IMDs did not differ between groups.

**Table 1. Demographic and clinical characteristics of the study cohort by origin.**

| Variable | Iraq<br>N = 188 | Northern Europe<br>N = 2207 | P value |
|---|---|---|---|
| Age at onset, years | 31.0 (8.0, 57.0) | 31.2 (2.0, 79.0) | NS |
| **Sex**<br>Women<br>Men | <br>109 (58.0)<br>79 (42.0) | <br>1529 (69.3)<br>678 (30.7) | <0.001 |
| Disease duration, years | 17.6 (10.4, 25.6) | 18.2 (10.7, 26.6) | NS |
| Disability score at onset* | 2.5 (0.0–7.0) | 2.0 (0.0–8.5) | NS |
| Patients treated with immunomodulatory drugs | 131 (69.7) | 1511 (68.5) | NS |

Categorical variables are shown as number and percent, and continuous variables are shown as median and interquartile range.

IQR, interquartile range, NS = not statistically significant.

*Disability was assessed by the Expanded Disability Status Scale (EDSS) score ranging from 0 to 10, with higher scores indicating greater disability.

## Long-term disability progression

During the follow-up period, 97 of 188 (51.6%) Iraqi origin relapsing-remitting MS patients and 975 of 2207 (44.2%) Northern Europe origin relapsing-remitting MS patients progressed to moderate disability, p = 0.0496. Progression to severe disability occurred in 50 of 188 (26.6%) Iraqi origin relapsing-remitting MS patients and 593 of 2207 (26.9%) of Northern Europe origin relapsing-remitting MS patients, p = 0.9353, Table 2.

Hazard ratios for moderate and severe disability progression calculated with multivariable Cox proportional-hazard models and adjusted for sex, disease duration, age at onset, and treatment with IMDs are presented in Tables 3 and 4, respectively. The risk for moderate disability progression was significantly higher (adj. hazard ratio, 1.323; 95% confidence interval, 1.049 to 1.668; p = 0.0183) for relapsing-remitting MS patients with an Iraqi origin, but not for severe disability progression (adj. hazard ratio, 1.311; 95% CI, 0.918–1.874, p = 0.136).

The cumulative incidence of progression to moderate and severe disability over time in relation to origin by Kaplan-Meier estimates and log-rank test are presented in Fig 1A and 1B, respectively.

## Transcriptional profile associated with worse clinical outcome in relapsing-remitting MS patients originated in Iraq

Transcriptional gene expression profile demonstrated 98 differentially expressed genes, 79 over-expressed and 19 under-expressed, (S1 Table), between relapsing-remitting MS patients originated in Iraq (n = 17, mean age 37.5±12.9 years, 12 women, disease duration 7.3±6.7 years, EDSS 2.7±1.7, median 2.5), and relapsing-remitting MS patients from Northern European origin (n = 34, age 39.9±13.7 years, 20 women, disease duration 12.2±9.9 years, EDSS 2.8 ±2.2, median 2.2). No statistically significant differences were found between the groups in relation to age and disability at time of blood sampling, but patients who originated in Iraq

**Table 2. Progression to moderate and severe disability by origin.**

| Disability progression | Iraq<br>N = 188<br>n (%) | Northern Europe<br>N = 2207<br>n (%) | Odds ratio<br>(95% confidence interval) | P value |
|---|---|---|---|---|
| **Moderate, n (%)** | 97 (51.6) | 975 (44.2) | 1.347 (1.042–1.815) | 0.0496 |
| **Severe, n (%)** | 50 (26.6) | 593 (26.9) | 0.986 (0.704–1.3812) | 0.9353 |

**Table 3. Adjusted hazard ratio for progression to moderate disability.**

| Parameter | Chi-squared P value | Hazard Ratio* | (95% confidence interval) |
|---|---|---|---|
| **Origin: Iraq vs Northern Europe** | 0.0183 | 1.323 | (1.049–1.668) |
| **Sex: Men vs women** | 0.3396 | 1.077 | (0.925–1.255) |
| **Disease duration, Years** | < .0001 | 1.035 | (1.026–1.043) |
| **Age at onset, Years** | < .0001 | 1.023 | (1.017–1.030) |

*Adjusted to sex, disease duration, age at onset, and treatment with IMDs.

had shorter disease duration (p = 0.04), and reached a similar level of disability within a shorter period, rendering the cohort perfect to study the molecular pathways that could explain the reasons for faster disability progression. Examining the differentially expressed genes using the Ingenuity database, showed enrichment of neuronal pathways among which are genes associated with morphology of axons (p = 1.5E-02), branching of neurites (p = 5.3E-03), proliferation of neocortical neurons (p = 7.0E-03), formation of myelin sheath (p = 8.4E-03), and cell survival of hippocampal neurons (p = 2.1E-02), (S2 Table). Specifically, genes associated with proliferation of neocortical neurons, morphology of axons and morphology of white matter, were down-regulated in relapsing-remitting MS patients who originated in Iraq, while genes associated with repair mechanisms related to formation of myelin sheath, myelination of axon bundle, binding of neurons and size of presynaptic terminals were up-regulated in relapsing-remitting MS patients who originated in Iraq, Fig 2.

## Discussion

The prevalence of MS among the Jewish Israeli population who originated in Iraq is relatively low, similarly to the prevalence of MS in non-Jewish Iraqi population and the low MS prevalence is in Middle Eastern countries as compared to Northern Europe [9,10]. Despite the low prevalence for MS, we noticed in our day-to-day clinical work, that Jewish relapsing-remitting MS patients who originated in Iraq progress fast to disability. In the current study we evaluated this clinical observation in a large cohort of Jewish relapsing-remitting MS patients that were either born in Iraq or both their parents were born in Iraq. Our findings indeed demonstrate that progression to moderate disability occurred faster in relapsing-remitting MS patients with an Iraqi origin than in relapsing-remitting MS patients with a Northern European origin.

Though the findings showed a non-statistical trend for faster progression to severe disability, we assume that with a larger sample size this trend would become also statistically significant.

The reasons for different pattern of MS progression in various populations are many and include different genetic background, environmental factors, and immune exposure to various

**Table 4. Adjusted hazard ratio for progression to severe disability.**

| Parameter | Chi-squared P value | Hazard Ratio* | (95% confidence interval) |
|---|---|---|---|
| **Origin: Iraq vs Northern Europe** | 0.1364 | 1.311 | (0.918–1.874) |
| **Sex: Men vs women** | 0.2541 | 1.139 | (0.911–1.423) |
| **Disease duration, Years** | < .0001 | 1.072 | (1.063–1.082) |
| **Age at onset, Years** | < .0001 | 1.040 | (1.030–1.051) |

*Adjusted to sex, disease duration, age at onset, and treatment with IMDs.

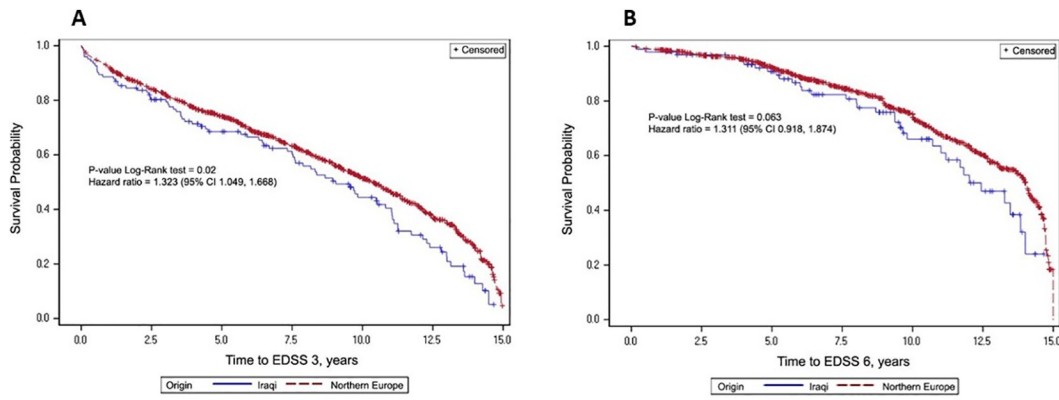

**Fig 1.** (A) Cumulative incidence of progression to moderate disability in relation to origin; (B) Cumulative incidence of progression to severe disability in relation to origin.

viral and bacterial antigens in early age [11–15]. The nature of people immigrating to Israel from either Iraq or Northern Europe was not affected by environmental or social variables that could be relevant to MS. We therefore further assessed the clinical difference in the rate of disease progression using gene expression arrays and identified several neuronal pathways that were enriched in relapsing-remitting MS patients who originated in Iraq. These include genes associated with neuronal growth, axonal integration, and myelin formation. Among these are DYRK1A (dual-specificity tyrosine phosphorylation regulated kinase 1A) gene that is mainly

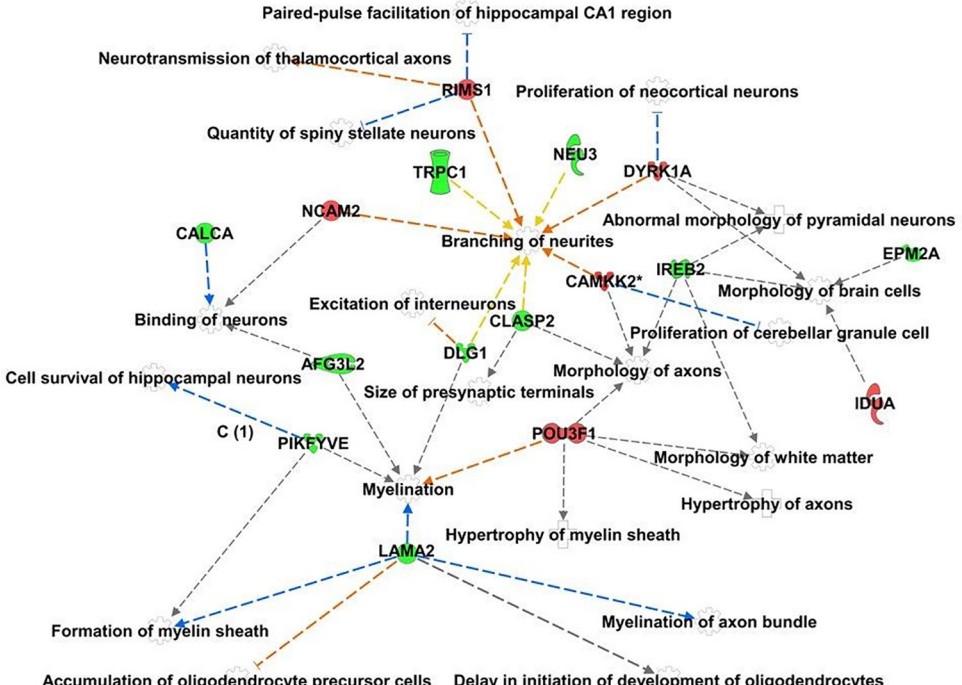

**Fig 2. Enriched neuronal genes and pathways in relapsing-remitting multiple sclerosis patients with an Iraqi origin.** The green color denotes over-expressed genes and the red color under-expressed genes in Iraqi origin MS patients.

found in growing dendrites and developing neuropil of differentiating pyramidal neocortical neurons [16]. Normal levels of DYRK1A are required to sustain neurogenesis in the adult brain, and down-expression of DYRK1A was suggested to suppress overproduction of astrocytes [17]. It is of note that DIRK1 also inhibits apoptosis by phosphorylating the NAD-dependent deacetylase Sirtuin-1 (SIRT1), an inhibitor of p53 [18], and therefore its down-regulation in Iraq-origin MS patients can further contribute to increase the inflammatory process in the brain and prevent neuronal repair. Similarly, NCAM2 (Cell adhesion molecule, neural, 2) is primarily expressed in the brain, where it stimulates neurite outgrowth and facilitates dendritic and axonal compartmentalization [19], and when suppressed the early transcriptional response associated with recovery and remyelination fails and does not materializes.

CLASP2 (cytoplasmic linker associated proteins), that was over-expressed in Iraq-origin MS patients, belongs to a heterogeneous family of plus-end tracking proteins that specifically accumulate at the growth cone extension of a regenerating neurite seeking its synaptic target. Therefore, CLASP2 is a key cytoskeletal effector during neuronal migration [20], and its overexpression in Iraq-origin MS patients may be a part of the modulation process to overcome the inflammatory insult. Similarly, over-expression of LAMA2 (Laminin α2 gene), that is known to regulate blood-brain barrier integrity and influence long-term synaptic plasticity [21], contributes to the on-going recovery processes. Neurite outgrowth and axon pathfinding are influenced by α2-containing laminins, and therefore, similarly to CLASP2, its over-expression in Iraq-origin MS patients suggest an augmented repair process during enhanced neurodegeneration. It was recently shown that central nervous system-resident pericytes respond to demyelination by proliferation and secretion of LAMA2 to enhance oligodendrocyte progenitor cells differentiation [22], and therefore its overexpression in Iraq-origin MS patients may further signify extended remyelination during the process of injury repair.

Although the gene expression findings are only suggestive as they are seen in peripheral blood mononuclear cells and therefore the relevance to the central nervous system is not direct, the relatively large number of genes in the enriched expression analysis merits further inspection. We intend in a future study to investigate brain and spinal cord MRI variables of Iraq-origin MS patients in relevance with the gene expression findings, and evaluate whether these patients will demonstrate more prominent changes of neuronal loss and brain atrophy as compared to the North-European originated patients.

To conclude, our findings suggest that in Iraq-origin relapsing-remitting MS patients the enhanced progression to disability signifies an interplay between over-expression of genes associated with inhibition of neuronal proliferation and down-expression of genes associated with myelination and neurite growth. This may lead to impaired recovery during the on-going demyelinating inflammatory insult and augment the process of neurodegeneration.

## Supporting information

**S1 Table. Differentially expressed genes (79 over-expressed and 19 under-expressed) in relapsing-remitting multiple sclerosis patients of Iraqi vs. northern European origin.** (DOCX)

**S2 Table. Enriched neuronal genes and pathways in relapsing-remitting multiple sclerosis patients with an Iraqi origin.** (DOCX)

## Author Contributions

**Conceptualization:** Anat Achiron.

**Data curation:** Gil Harari, Mark Dolev.

**Formal analysis:** Gil Harari.

**Methodology:** Michael Gurevich, Mark Dolev, Rina Zilkha Falb, Anat Achiron.

**Software:** Gil Harari, Anat Achiron.

**Writing – original draft:** Gil Harari, Anat Achiron.

**Writing – review & editing:** Gil Harari, Michael Gurevich, Mark Dolev, Rina Zilkha Falb, Anat Achiron.

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
