## [Decision Letter · Decision Letter 0]

13 Oct 2022

PONE-D-22-07527

Faster progression to multiple sclerosis disability is linked to neuronal pathways associated with neurodegeneration: An ethnicity study

PLOS ONE

Dear Dr. Harari

Thank you for submitting your manuscript to PLOS ONE. After careful consideration, we feel that it has merit but does not fully meet PLOS ONE’s publication criteria as it currently stands. Therefore, we invite you to submit a revised version of the manuscript that addresses the points raised during the review process.

Please submit your revised manuscript within 6 weeks. If you will need more time than this to complete your revisions, please reply to this message or contact the journal office at plosone@plos.org. Please include the following items when submitting your revised manuscript:

We look forward to receiving your revised manuscript.

Kind regards,

Alexander Klistorner

Academic Editor

PLOS ONE

“I have read the journal's policy and the authors of this manuscript have the following competing interests: Anat Achiron served on the scientific advisory board and/or received speaker honoraria/consulting from Biogen, BMS, Merck, Novartis, Roche and Sanofi, received grant/research support from Biogen, BMS, Merck, Roche and Sanofi and received institutional support from Sackler School of Medicine, Tel-Aviv University for Autoimmune Diseases research. All other authors have no competing interests.”

Reviewers' comments:

Reviewer's Responses to Questions

**Comments to the Author**

1. Is the manuscript technically sound, and do the data support the conclusions?

Reviewer #1: Yes

Reviewer #2: Partly

2. Has the statistical analysis been performed appropriately and rigorously? 

Reviewer #1: Yes

Reviewer #2: Yes

3. Have the authors made all data underlying the findings in their manuscript fully available?

Reviewer #1: No

Reviewer #2: Yes

4. Is the manuscript presented in an intelligible fashion and written in standard English?

Reviewer #1: Yes

Reviewer #2: Yes

5. Review Comments to the Author

Reviewer #1: The manuscript has been conducted in an admirable methodology as regards to the genetic study, yet there are certain issues that need to be fully elucidated for adopting the proposed generalizability of study findings: The radiologic progression findings for the patients (including but not limited to: MRI demyelinating patch count progression, the black hole count progression, the cerebral volume regression), and the +Ve consanguinity history in the study participants [which is an important possible confounder]. In addition, it has been noted that, in many MS patients of Middle East origin, the incidence of seizures is more as compared to those of Northern European origin.

Reviewer #2: Here, the authors describe a retrospective cohort study in Israel evaluating disability progression in 188 RRMS of Iraqi descent compared to 2,207 MS cases of northern European background. Over the periods of follow-up, a larger proportion of those from Iraq developed moderate disability, this borderline significant (OR=1.35, p=0.049). Similar results were seen evaluating by survival analysis (aHR=1.32, p=0.02). A similar association was seen for risk of developing severe disability was seen but did not reach significance (aHR=1.31, p=0.063). Evaluating PBC gene expression array data, the Iraqi cases showed enriched expression in neuronal pathway genes. The authors conclude this expression data suggests a mechanism underlying a differential disability progression in persons of Iraqi descent acting via increased neurodegeneration.

The fraught international relationships of countries like Iraq with Israel makes the comparison of this population with immigrants from Europe a bit complicated. The nature of people immigrating to Israel from these countries is likely to differ in ways that may be relevant to MS risk, e.g., SES, sociopolitical factors, etc. I think this may merit comment. I am trying to phrase this in a way that is not insensitive or discriminatory and hopefully I have not come across as such. I merely mean to state that because Iraq and other Muslim-majority countries have an acrimonious history with Israel, sometimes including the nations going to war, this may mean that people from Iraq may be differently disposed or able to immigrate to Israel, as compared to those from northern European countries which has not had such issues in their international relations. This has bearing then in the comparability of these patient groups, particularly in the interpretation of observed differences between those of Iraqi and European background as being due to genetic factors, rather than environmental, lifestyle, and other exposures which may differ between the groups. If the authors might please speak to this, I think it relevant.

Further to this, I wonder whether there are immigrants from other countries in the region besides Iraq which might be available for comparison. The authors justify the selection of Iraqis for evaluation because they have observed Iraqi RRMS patients in their clinic to have a worse disease course. It would be supportive to the authors’ interpretation of the results as being something genetically specific to the Iraqis if immigrants from other countries in the region could also be assessed and see whether they also show any differences in disability progression. I appreciate this is a significant amount of new data collection and analysis, however, so if the authors are not able to do this, they could just suggest this as worthy of future investigation.

I don’t see how enriched expression of genes involved in neuronal function, including neuronal proliferation and structure and myelin sheath, is indicative of a pathway of increased neurodegeneration. It becomes apparent in the body text that proliferation and morphology loci were downregulated in the Iraqi patients, while myelination and myelin repair and synapse loci were upregulated. While the neuronal proliferation downregulation might be a negative, I am uncertain that the morphology downregulation is clearly deleterious. Also, myelination and myelin repair loci being upregulated would seem a good thing. This all of course presumes that the expression seen in PBMCs is relevant to expression that may be occurring in the CNS. I think these expression data are suggestive, albeit a bit mixed, and so the results should be framed a bit more conservatively, as they don’t clearly indicate a neurodegeneration profile and in any case the extrapolation of these results to long-term disability progression is uncertain.

Further to this, please acknowledge the limitation in the Discussion that expression in PBMCs may not be reflective of the expression profile in the CNS. This goes without saying but the authors are framing results for expression of genes involved in CNS function and repair in PBMCs, so it would be good to be explicit and acknowledge this limitation and the need to assess whether such differences in expression are also seen in the CNS.

Please consider evaluating disability as a continuous term and looking at rate of change. Also, please consider looking at time to event for reaching EDSS 4 and 6, as these are more typical cutpoints for disability than moderate and severe. This would better support the framing of conclusions as Iraqis having faster disability progression.

The framing of results as Iraqi-descent patients having a greater risk of moderate but not severe disability is not really appropriate. The magnitudes of association are statistically identical and the level of statistical significance is not that different either. Just because one falls below p=0.05 but not the other is not reason to disregard the association for severe disability. The moderate disability association can certainly be framed as having a higher level of statistical evidence but it’s not right to say there was no association for severe disability.

The final paragraph of the Discussion states that there is an overexpression of genes involved in repressing neuronal proliferation and downexpression of genes involved in myelination. This conflicts with results presented earlier, however, where it says that genes involved in neuronal proliferation were downregulated and that those for myelination and myelin repair were upregulated. Please assess and remedy any discrepancy.

Abstract:

- Please state where the study was conducted.

- Please specify that the 2,207 cases are also RRMS.

- Please specify how disability was assessed and what defines moderate and severe disability.

- Please specify what models are adjusted for. Also, for internal comparability, please specify what the association by logistic regression adjusting for these factors is.

- Please specify that the gene expression enrichment was different for Iraqi vs European cases and consider specify the numbers of upregulated and downregulated genes.

Introduction:

- Consider different word choice for “the Orient”.

Methods:

- Please specify whether the patients from European countries are also Jewish? It is stated that the Iraqis are so I reckon this is relevant.

- In the inclusion criteria, it is stated that Iraqi-descent patients were included but earlier in the Methods it states they were Jewish. Please ensure consistency.

- Further to this, the inclusion criteria states that the European patients had to be Ashkenazi Jews. Were the Iraqis also Ashkenzai or were they a mixture of Jewish types? Is this likely to have relevance for analyses? If so, please speak to this in the Discussion. The authors might also consider, sample allowing, to do a sensitivity analysis constraining the Iraqis to Ashkenazi Jews and seeing whether disability progression differs between them and the European group.

- Were analyses adjusted for DMTs considered? If so and results did not materially differ, this can just be stated.

Formatting/other:

- Please use sex and male/female terminology, rather than gender and men/women.

6. PLOS authors have the option to publish the peer review history of their article (what does this mean?). If published, this will include your full peer review and any attached files.

Reviewer #1: **Yes: **Mohamed Mostafa

Reviewer #2: **Yes: **Steve Simpson-Yap

---

## [Author Response · Author response to Decision Letter 0]

17 Dec 2022

Dear Editor,

Enclosed please find the revised version of our manuscript entitled: Faster progression to multiple sclerosis disability is linked to neuronal pathways associated with neurodegeneration: An ethnicity study.

We have addressed all the points raised during the review process.

Please find our responses to the Reviewers comments. We hope in that the revised corrected manuscript will be appropriate for publication in PlosOne.

Conflicts of Interest and Disclosure: Anat Achiron served on the scientific advisory board and/or received speaker honoraria/consulting from Biogen, BMS, Merck, Novartis, Roche and Sanofi, received grant/research support from Biogen, BMS, Merck, Roche and Sanofi and received institutional support from Sackler School of Medicine, Tel-Aviv University for Autoimmune Diseases research. All other authors have no competing interests. This does not alter our adherence to PLOS ONE policies on sharing data and materials

With respects,

Dr Gil Harar, PhD

---

## [Editor Report · Decision Letter 1]

2 Jan 2023

Faster progression to multiple sclerosis disability is linked to neuronal pathways associated with neurodegeneration: An ethnicity study

PONE-D-22-07527R1

Dear Dr. Harari,

We’re pleased to inform you that your manuscript has been judged scientifically suitable for publication and will be formally accepted for publication once it meets all outstanding technical requirements.

Kind regards,

Alexander Klistorner

Academic Editor

PLOS ONE
---

## [Editor Report · Acceptance letter]

27 Jan 2023

PONE-D-22-07527R1 

Faster progression to multiple sclerosis disability is linked to neuronal pathways associated with neurodegeneration: An ethnicity study 

Dear Dr. Harari:

I'm pleased to inform you that your manuscript has been deemed suitable for publication in PLOS ONE. Congratulations! Your manuscript is now with our production department. 

Kind regards, 

on behalf of

Dr. Alexander Klistorner 

Academic Editor

PLOS ONE